# Association of DNA Methylation of the *NLRP3* Gene with Changes in Cortical Thickness in Major Depressive Disorder

**DOI:** 10.3390/ijms23105768

**Published:** 2022-05-21

**Authors:** Kyu-Man Han, Kwan Woo Choi, Aram Kim, Wooyoung Kang, Youbin Kang, Woo-Suk Tae, Mi-Ryung Han, Byung-Joo Ham

**Affiliations:** 1Department of Psychiatry, Korea University Anam Hospital, Korea University College of Medicine, Seoul 02841, Korea; han272@korea.ac.kr (K.-M.H.); choikwanwoo@gmail.com (K.W.C.); 2Department of Biomedical Sciences, Korea University College of Medicine, Seoul 02841, Korea; chloe_ark@korea.ac.kr (A.K.); wykang@korea.ac.kr (W.K.); youbin_kang@daum.net (Y.K.); 3Brain Convergence Research Center, Korea University College of Medicine, Seoul 02841, Korea; woostae@korea.ac.kr; 4Division of Life Sciences, College of Life Sciences and Bioengineering, Incheon National University, Incheon 22012, Korea

**Keywords:** neuroinflammation, DNA methylation, epigenetics, NLRP3, major depressive disorder, depression, magnetic resonance image, neuroimaging, cortical thickness

## Abstract

The Nod-like receptor pyrin containing 3 (NLRP3) inflammasome has been reported to be a convergent point linking the peripheral immune response induced by psychological stress and neuroinflammatory processes in the brain. We aimed to identify differences in the methylation profiles of the *NLRP3* gene between major depressive disorder (MDD) patients and healthy controls (HCs). We also investigated the correlation of the methylation score of loci in *NLRP3* with cortical thickness in the MDD group using magnetic resonance imaging (MRI) data. A total of 220 patients with MDD and 82 HCs were included in the study, and genome-wide DNA methylation profiling of the *NLRP3* gene was performed. Among the total sample, 88 patients with MDD and 74 HCs underwent T1-weighted structural MRI and were included in the neuroimaging–methylation analysis. We identified five significant differentially methylated positions (DMPs) in *NLRP3*. In the MDD group, the methylation scores of cg18793688 and cg09418290 showed significant positive or negative correlations with cortical thickness in the occipital, parietal, temporal, and frontal regions, which showed significant differences in cortical thickness between the MDD and HC groups. Our findings suggest that *NLRP3* DNA methylation may predispose to depression-related brain structural changes by increasing NLRP3 inflammasome-related neuroinflammatory processes in MDD.

## 1. Introduction

Major depressive disorder (MDD) is characterized by multifaceted neurobiological etiologies, such as genetic predisposition, disturbances in monoamine and stress hormone systems, and functional and structural alterations in brain networks [1]. Among these, the contribution of neural–immune interactions to the pathogenesis of MDD has been the focus of attention for the past three decades [2]. Preclinical studies have supported this hypothesis by demonstrating that immune challenges can induce depression-like behavior in animal models [3]. Recent meta-analyses from human studies support this hypothesis by identifying that the levels of peripheral inflammatory markers, such as C-reactive protein (CRP), interleukin-1 (IL-1), IL-2, IL-6, interferon-γ (IFN-γ), and tumor necrosis factor-α, were elevated in MDD patients compared to healthy controls (HCs) [4,5,6].

Heritability estimates range from only 40–50% for MDD [7], and genetic variation alone is not completely attributable for the risk of developing MDD. Environmental factors, such as adverse life events and psychological or physical stress, are known to be the main risk factors for MDD and are associated with inflammatory processes [6,8,9]. Epigenetic modifications of genes associated with inflammatory processes could reveal chronic pathophysiological alterations in MDD patients [10,11]. Interestingly, a growing body of evidence has identified that epigenetic regulation of inflammation-related genes, such as DNA methylation, is deeply involved in the pathophysiology of MDD. A recent genome-wide methylation study found that the methylation status of the inflammation-related gene pathway was significantly associated with the recurrence of MDD [12]. This finding is supported by a meta-analysis of seven genome-wide association studies, which found that inflammation-related genes (e.g., *LRFN5*) and genetic pathways involved in cytokine and immune responses were associated with MDD [13]. A recent study also reported that the comethylation module in the weighted gene comethylation network analysis was associated with a history of depression and elevated IL-6 serum levels [11]. These findings suggest that predisposition to MDD is affected by an individual’s inflammation-related genetic signature, which could be modulated by epigenetic mechanisms such as DNA methylation.

The most proximal part of the neuroinflammation-related pathophysiological processes in MDD may be functional and structural alterations of neural circuits resulting from this neurotoxic cascade [14]. However, little is understood at the clinical research level regarding the relationship between inflammatory state and structural and functional brain changes in MDD. Furthermore, given the importance of epigenetic dysregulation of inflammation-related genes in the pathophysiology of MDD [11], it is plausible to investigate the relationship between methylomic profiles and neuroimaging markers in MDD.

The Nod-like receptor pyrin containing 3 (NLRP3) inflammasome has been reported to be a convergent point linking the psychological stress-induced peripheral immune response and the neuroinflammatory processes in the brain [15]. Preclinical studies have reported that chronic stress-induced depressive behavior was associated with the activation of the NLRP3 inflammasome in rats [16,17], and some human studies have shown that MDD patients demonstrated increased levels of *NLRP3* mRNA compared to HCs [18,19]. These findings support the hypothesis that *NLRP3* and its genetic regulation may be key mediators in the pathway between chronic stress and the neuroinflammatory processes that concern the pathophysiology of MDD. Furthermore, a growing number of studies have shown that DNA methylation of the *NLRP3* gene plays a critical role in the expression, assembly, and activation of the NLRP3 inflammasome [3].

Therefore, herein, we aimed to identify differences in the methylomic profiles of the *NLRP3* gene between MDD patients and HCs to examine the contribution of the epigenetic regulation of *NLRP3* to the pathophysiology of MDD. We also investigated the correlation of methylation score of loci in *NLRP3* to cortical thickness in the MDD group using T1-weighted magnetic resonance imaging (MRI) data to elucidate the relationship between epigenetic regulation of *NLRP3* and structural brain changes in MDD. Our a priori hypothesis was as follows: the MDD group will show a significantly different methylation signature of *NLRP3* compared to HCs, which will significantly correlate with cortical thickness in the MDD group. To examine our a priori hypothesis, we performed an epigenomic analysis focusing on the *NLRP3* gene and a cortical thickness analysis.

## 2. Results

### 2.1. Differential Methylation Analysis

A total of 220 MDD patients and 82 HCs (total sample) were included in the differential methylation analysis (Appendix A). For the neuroimaging analysis, 88 MDD patients and 74 HCs were included (Table 1). No significant differences were observed between the groups in age, sex, or education level (all *p* > 0.1), except for the HDRS score (t = 18.807, *p* < 0.001).

After comprehensive quality-control procedures, 753,531 CpGs were retained and used for DMP analysis. We identified five significant DMPs (*p* < 0.05, Q = 0.15) in the *NLRP3* gene, which included DMPs with absolute changes in average methylation (Δβ) ranging from 0.001–0.028 (Table 2). Three DMPs were located in gene bodies, one in a region up to 200 bp upstream from the transcription start site (TSS), and one in the 3′ untranslated region (UTR).

### 2.2. Cortical Thickness Analysis

In the comparison of cortical thickness between the groups, the MDD group showed 15 cortical regions with significantly reduced thickness and 16 cortical regions with significantly increased thickness compared to the HC group after Bonferroni correction: *p* < 4.11 × 10^−5^ (Table 3).

Cortical regions with significantly lower thickness in the MDD group included the ventral posterior cingulate gyrus, superior parietal lobule, middle occipital gyrus, lateral occipitotemporal gyrus, middle temporal gyrus, planum polare, right precentral gyrus, subcallosal gyrus, and short insula gyrus (Table 3). Cortical regions with significantly greater thickness in the MDD group included the posterior mid-cingulate gyrus, cuneus, superior occipital gyrus, lingual gyrus, postcentral gyrus, lateral superior temporal gyrus, left precuneus, paracentral lobule, right frontomarginal gyrus, and planum temporale (Table 3).

### 2.3. Correlation between DNA Methylation and Cortical Thickness

We performed a correlation analysis using the methylation score data of five significant DMPs and cortical thickness from the neuroimaging sample. In the correlation analyses, we found that the methylation scores of cg18793688 and cg09418290 in *NLRP3* were significantly correlated with cortical thickness in MDD patients (Table 4, Figure 1). In the MDD group, the methylation score of cg18793688 showed an inverse correlation with the thickness of the superior frontal gyrus (left: r = −0.359, P_corr_ (corrected *p*-value) = 0.038; right: r = −0.388, P_corr_ = 0.037), cuneus (left: r = −0.364, P_corr_ = 0.038; right: r = −0.378, P_corr_ = 0.037), and lingual gyrus (left: r = −0.383, P_corr_ = 0.037; right: r = −0.447, P_corr_ = 0.011) in both hemispheres, and the right postcentral gyrus (r = −0.394, P_corr_ = 0.037), as shown in Table 3. There was a positive correlation between the methylation score of cg18793688 and the thickness of the right middle temporal gyrus in the MDD group (r = 0.362, P_corr_ = 0.038) in the right hemisphere. Among these cortical regions, both the cuneus and lingual gyrus and the right postcentral gyrus showed significantly greater thickness, while the right middle temporal gyrus showed significantly lower thickness in the MDD group than in the HC group (Table 3). We did not observe any significant association between the variable of taking psychotropic medication and the methylation scores of the five significant DMPs. Furthermore, there was no significant difference in cortical thickness between drug-naïve patients and those taking psychotropic medications in the MDD group.

The methylation score of cg09418290 also demonstrated a positive correlation with cortical thickness of the left superior occipital gyrus (r = 0.369, P_corr_ = 0.037) and the right planum temporale (r = 0.373, P_corr_ = 0.037) in the MDD group (Table 4). These cortical regions showed significantly greater thickness in the MDD group than in the HC group (Table 3). In the HC group, there was no significant correlation between the methylation scores of any DMPs and cortical thickness.

## 3. Discussion

To our knowledge, this is the first study to investigate the association between DNA methylation levels of the *NLRP3* gene and changes in cortical thickness in MDD patients. This study has three important findings. First, we identified several significant DMPs in *NLRP3* in MDD patients compared to HCs. Second, we showed that MDD patients had significant correlations between the DNA methylation levels of *NLRP3* and the cortical thickness of related regions, which were significantly altered in MDD patients compared to HCs. Our findings are based on a relatively medium-to-large sample size, along with conservative multiple-comparison correction and adjustment for potential confounders.

Herein, we found five differentially methylated CpG sites in the *NLRP3* gene, located in the gene body, TSS200, and 3′UTR regions, in the comparison between the MDD and HC groups with a comprehensive investigation of 753,531 CpGs. *NLRP3* is a major component of the NLRP3 inflammasome, which is an intracellular multiprotein complex composed of NLRP3 as a sensor, an apoptosis-associated speck-like protein containing a caspase activation and recruitment domain as an adaptor, and caspase-1 as an effector protein [20,21]. The NLRP3 inflammasome machinery plays an important role in innate immunity by activating caspase-1, furthering the maturation and secretion of IL-1β and IL-18 and producing the corresponding mature cytokines [22]. The expression and activation of the NLRP3 inflammasome is critical for the development of stress-induced depressive-like behaviors [23,24]. The NLRP3 inflammasome complex, which is associated with damage-associated molecular patterns (DAMPs), can detect various danger signals and produce accompanying immune-inflammatory reactions, the processing and release of IL-1β and IL-18, and pathogen-associated molecular patterns (PAMPs) [23,24,25]. In contrast to PAMPs, which are related to foreign microbes or viruses, DAMP-related signaling occurs in the absence of pathogens or sterile inflammation, such as psychological or physical stressors [26]. The DMPs we discovered in NLRP3 may be critical mediators by which psychological stressors can contribute to the development of MDD as a neuroinflammatory illness [16,23].

Our results discovered several significant alterations in cortical thickness in MDD patients compared to HCs. Thinner cortical regions, such as the posterior cingulate gyrus [27], middle temporal gyrus [27,28], and short insular gyrus [27], in MDD patients are consistent with previous meta-analytic studies, although a thicker lingual gyrus in both hemispheres in MDD patients is quite contrary to the findings of other meta-analyses and similar studies [29,30]. Furthermore, we found additional alterations in cortical thickness in several regions, including the cortical areas of the pre- and postcentral gyrus [30], occipital (middle and superior occipital gyrus), lingual gyrus [31], cuneus [32], and parietal (superior parietal and precuneus) areas, which have been known to be associated with MDD. These findings are consistent with those of a recent meta-analysis from a large-size ENGMA consortium, which showed significantly reduced cortical thickness in the prefrontal regions, anterior and posterior cingulate gyrus, precuneus, cuneus, and precentral gyrus in MDD patients compared to HCs [27]. Particularly, cortical thinning in the right middle frontal gyrus, which has a positive correlation with the cg18793688 methylation score, was inconsistent with that in a previous voxel-based morphometry study, which reported a gray-matter volume reduction in the middle temporal gyrus and PFC in the right hemisphere in MDD compared to that in HCs [33].

In the present study, the DNA methylation levels of *NLRP3* were significantly correlated with cortical thickness in the various regions of the brain that showed significantly greater or lower cortical thickness in MDD patients compared to HCs, such as the bilateral lingual gyrus, the left superior occipital gyrus, the right postcentral gyrus, the right middle temporal gyrus, the right planum temporale, and the left superior frontal gyrus. Our findings implicated NLRP3 inflammasome-related neuroinflammatory processes in MDD, consistent with previous preclinical and clinical studies. Preclinically, Iwata et al. (2016) reported that Nlrp3-null mice displayed decreased anxiety and anhedonic behaviors even under basal, unstressed conditions and were resilient to the behavioral deficits caused by chronic unpredictable stress exposure, compared to wild-type mice. Pan et al. [17] also showed that chronic unpredictable stress caused microglial *NLRP3* overexpression and impairment of astrocytic function in the prefrontal cortex (PFC) of rats with chronic unpredictable stress, which were reversible through long-term fluoxetine treatment. Several postmortem studies have shown altered NLRP3 and complex I levels in the frontal cortical tissue of MDD patients [15,34,35]. Another postmortem evaluation of brain tissue revealed that microglia exhibit a reactive morphology that was associated with a higher expression of NLRP3 inflammasome-related neurotoxic metabolite—quinolinic acid—in the subgenual anterior cingulate cortex and the anterior midcingulate cortex in MDD patients [36]. Recent clinical studies have reported elevated *NLRP3* mRNA expression and NLRP3 protein levels in peripheral blood mononuclear cells in MDD patients [18,19,37]. Previous findings suggest that immune dysregulation, specifically of inflammatory processes, is intricately intertwined with MDD [38,39,40,41]. Impaired immune systems can trigger cascades of neurotoxic processes by affecting the metabolism of norepinephrine, 5-hydroxytryptamine, dopamine, and neuroendocrine function [42,43], resulting in an increased production of neurotoxic kynurenine pathway metabolites (e.g., quinolinic acid, 3-hydroxy-kyunurenine), altered glial cell activities, and impaired neurogenesis [2,40,44].

In the present study, the DNA methylation of NLRP3 was strongly associated with changes in cortical thickness in the occipital, parietal, temporal, and frontal regions in the MDD group. Although no clinical studies have investigated the association between NLRP3 or its methylation changes and brain structural alterations in MDD, several neuroimaging studies have identified that increased peripheral inflammatory markers are associated with brain structural changes in MDD [45,46,47]. For example, Opel et al. [46] found that the MDD group showed significantly higher serum CRP levels compared to HCs, which was negatively correlated with gray-matter volumes in the PFC in the MDD group. In particular, regarding the DNA methylation of inflammatory genes, only one study by Green et al. showed a significant association between a DNA methylation score for CRP and entorhinal volume and diffuse white-matter disruption in MDD patients [45]. Our previous study also reported a significant inverse correlation between serum levels of FAM19A5, a chemokine-like peptide that reflects reactive astrogliosis and inflammatory activation in the brain, and cortical thickness in the PFC, precuneus, cuneus, and posterior cingulate gyrus, which showed reduced thickness in MDD patients compared to HCs [47]. Our findings may provide a likely explanation that DNA methylation changes in *NLRP3* may affect predisposition to NLRP3 inflammasome-related neuroinflammatory processes in response to psychological stressors, which in turn may lead to structural brain changes through the neurotoxic metabolite synthesis of the kynurenine pathway and glutamate-mediated neuronal excitotoxicity [2,15]. This may eventually result in depression-related brain network dysfunction. However, since there have been limited neuroimaging studies on *NLRP3* or its DNA methylation changes, more studies are needed to clarify how NLRP3 inflammasome-related neuroinflammatory processes could affect brain structural changes in MDD.

This study has some limitations. First, this study used a cross-sectional design. Furthermore, we did not explore potential psychosocial stressors (e.g., early life adversity), which could affect the relationship between NLRP3 methylation and brain structural alterations in MDD. Therefore, we could not clearly delineate how psychosocial stressors could lead to changes in cortical thickness through DNA methylation changes in NLRP3 in a longitudinal manner. Second, we recruited MDD patients and HCs who were significantly different from each other in terms of their mean age. The prototypic lifespan trajectory includes a steep age-related decrease in childhood and adolescence, followed by a mild monotonic thinning from early adulthood and, in many regions, an accelerated thinning from approximately the seventh decade of life [48]. Although we adjusted for age differences in all neuroimaging analyses by including age as a covariate, the difference in mean age between the two groups may have affected our findings. Finally, we did not include variables such as body mass index, cigarette smoking, or alcohol consumption, which can affect DNA methylation status [49], in the DNA methylation analysis, which may have affected our results.

To our knowledge, this is the first study to report the association between several DNA methylation sites in the *NLRP3* gene and cortical thickness in the occipital region, as well as the frontal, parietal, and temporal regions of the brain in MDD patients. A growing body of evidence has shown that methylation signatures can provide a deep insight into the interplay between the psychosocial environment and disease trajectory of MDD [11,12]. Our findings suggest that the neuroinflammation-related methylomic profiles of individuals may provide useful information about the neurobiological processes among psychosocial stress, heightened inflammation states, structural alterations of the brain, and predisposition to MDD [2,50,51]. This could be helpful in the realization of the concept of “evidence-based psychiatry” in MDD [52]. Future studies are needed to confirm the utility of DNA methylation of the ***NLRP3*** gene as a biomarker for neuroinflammatory changes and pathophysiological processes in MDD.

## 4. Materials and Methods

### 4.1. Participants

The present study included 220 MDD patients and 82 HCs. Patients were recruited from the outpatient psychiatric clinic of Korea University Anam Hospital in Seoul, Republic of Korea, between February 2010 and June 2018. The patients included in the study were adults aged 19–65 years. The diagnosis of MDD was confirmed by two board-certified psychiatrists (K.M. Han & B.J. Ham) using a structured clinical interview for the Diagnostic and Statistical Manual of Mental Disorders, Fourth Edition (DSM-IV) Axis I Disorders (SCID-I). Only patients whose diagnoses were confirmed by both psychiatrists were enrolled. The exclusion criteria were as follows: (i) comorbidity of any other major psychiatric disorders (including personality and substance use disorders); (ii) MDD with psychotic features; (iii) acute suicidal or homicidal patients requiring inpatient treatment; (iv) history of major medical conditions; (v) primary neurological illness; and (vi) any contraindication for MRI. The duration of illness was assessed using life charts. Eighty-two HCs aged 19–65 years were recruited from the community using advertisements. Two psychiatrists evaluated the HCs and confirmed the absence of current or previous psychiatric disorders. The same exclusion criteria were applied to the HC group. The severity of depressive symptoms in all subjects was assessed using the 17-item Hamilton Depression Rating Scale (HDRS) [53]. Among the 220 MDD patients and 82 HCs, 88 patients and 74 HCs underwent T1-weighted MR imaging and were included in the neuroimaging analysis (i.e., neuroimaging sample). In the neuroimaging sample, we confirmed that all participants were right-handed according to the Edinburgh handedness test [54]. The demographic and clinical characteristics of the subsample in the neuroimaging analysis are presented in Table 1. At the time of study enrollment, 56 patients were being treated with antidepressants and 32 patients were drug-naïve within the neuroimaging sample (Table 1).

### 4.2. Methylomic Profiling of NLRP3 Gene

Genome-wide DNA methylation profiling of NLRP3 was performed using the Illumina Infinium Methylation EPIC Beadchip array (Illumina, San Diego, CA, USA), which covers 850,000 bp cytosine–phosphate–guanine (CpG) sites, including the entire sample (i.e., 220 patients with MDD and 82 HCs). The methylation beta (β) value, which is the ratio between methylated probe intensity and total probe intensities (sum of methylated and unmethylated probe intensities) and ranges from 0 to 1, was used as the proportion of methylation.

Genomic DNA was quantified using a NanoDrop^®^ ND-1000 UV–Vis Spectrophotometer (NanoDrop Technologies, Wilmington, DE, USA), and intact genomic DNA was diluted to 50 ng/μL concentration based on Quant-iT Picogreen quantitation (Invitrogen, Carlsbad, CA, USA). Bisulfite conversion of 500 ng of genomic DNA was performed using the Zymo EZ DNA methylation kit (Zymo Research, Irvine, CA, USA) according to the manufacturer’s protocol. The bisulfite-treated samples were then amplified, fragmented, purified, and hybridized onto the EPIC Beadchip according to the manufacturer’s standard protocol. The arrays were washed and scanned using the Illumina iScan platform.

### 4.3. Genome-Wide DNA Methylation Analysis of the NRLP3 Gene

The Illumina’s Genome Studio software was used to extract the signal intensities for each probe. Pre-processing of raw data was conducted using the ChIP R package [55]. We used Illumina Intensity Data files as input using “minfi” quality control and normalization options [56]. Samples were removed if they showed lower median intensities in methylated or unmethylated signals or if they failed to cluster with others based on all probes using hierarchical clustering. Probes with a signal detection *p* > 0.01 in at least one sample, probes with <3 beads in >5% of samples, non-CpG probes, cross-reactive probes, sex chromosome probes, and probes that hybridize to single nucleotide polymorphism sites were removed. Beta-mixture quantile normalization was used to adjust the bias introduced by the Infinium Type II probes. A singular value decomposition analysis was performed to detect technical batches and covariates, and batch effects were corrected before differential methylation analysis. Beta (β) values were generated as the ratio of methylated signal to the sum of methylated and unmethylated signals at each CpG site.

Differentially methylated positions (DMPs) on the NLRP3 gene were identified between the MDD and HC groups using linear regression methods, from the limma with age and sex as covariates (Ritchie et al., 2015). To correct for multiple testing, we used the Benjamini–Hochberg procedure in limma, which is based on false discovery rate (FDR) calculations [57].

### 4.4. MRI Data Acquisition

We obtained T1-weighted images of the participants using a 3.0-Tesla TrioTM whole-body imaging system (Siemens Healthcare GmbH, Erlangen, Germany) at the Korea University MRI Center. T1-weighted images were acquired parallel to the anterior-commissure–posterior-commissure line using the 3D T1-weighted magnetization-prepared rapid gradient-echo (MP-RAGE) sequence with the following parameters: repetition time (TR), 1900 ms; echo time (TE), 2.6 ms; field of view, 220 mm; matrix size, 256 × 256; slice thickness, 1 mm; number of coronal slices, 176 (without gap); voxel size, 0.86 × 0.86 × 1 mm^3^; flip angle, 16° flip angle; and number of excitations, 1.

### 4.5. Neuroimage Processing

We calculated the gray-matter thickness of each cortical region from T1 image data using the automated procedures implemented in the FreeSurfer 6.0 version (Laboratory for Computational Neuroimaging, Athinoula A. Martinos Center for Biomedical Imaging, Charlestown, MA, USA; http://surfer.nmr.mgh.harvard.edu (accessed on 1 April 2022)), which provides a 3D model of the cortical surface reconstructions [47]. Detailed technical information about the FreeSurfer procedure has been previously described [58,59,60,61,62,63]. The processes involved in the calculation of the cortical thickness in FreeSurfer were described in our previous studies [47,64,65,66]. Cortical reconstructions for the automatic segmentation of gray/white-matter boundaries were inspected, and data with inaccuracies were discarded. Cortical thickness was determined by calculating the shortest distance between the gray/white matter and pial surfaces using the surface deformation algorithm, measured in millimeters (mm) [47]. Smoothing of the cortical map was performed using a Gaussian kernel with a full width at half maximum of 20 mm for all cortical analyses [47]. Automatic parcellation per hemisphere into 74 cortical gyri and sulci was performed according to the atlas by Destrieux et al. [67], and the cortical thickness of each region was calculated. In the present study, we used the automatically calculated gray-matter thickness value of the 38 cortical gyri in each hemisphere, as described in our previous study [65,66].

### 4.6. Statistical Methods for Neuroimaging–DNA Methylation Correlation Analysis

First, we compared the cortical thickness of 76 cortical gyri in the bilateral hemispheres between the MDD and HC groups using a one-way analysis of covariance, including diagnosis (MDD vs. HC) as an independent variable, 76 extracted values of cortical thickness as dependent variables, and age, sex, education level, and total intracranial cavity volume (TICV), which were manually measured as previously described, as covariates [68]. For the multiple-comparison correction, we applied the Bonferroni correction to the cortical thickness comparison analysis: *p* < 4.11 × 10^−5^ (=0.05/76 cortical regions). Second, we performed a correlation analysis between the cortical thickness and the DNA methylation score of the CpG sites using Pearson’s partial correlation analysis for both groups. In the HC group analysis, age, sex, education level, and TICV were included as covariates. The correlation analysis in the MDD group included the HDRS score, illness duration, and medication (drug-naïve patients, coded as 0; patients taking antidepressants, coded as 1) as additional covariates. For neuroimaging–methylation analysis, FDR correction was applied to each analysis for multiple-comparison correction (q < 0.05).

## Figures and Tables

**Figure 1 ijms-23-05768-f001:**
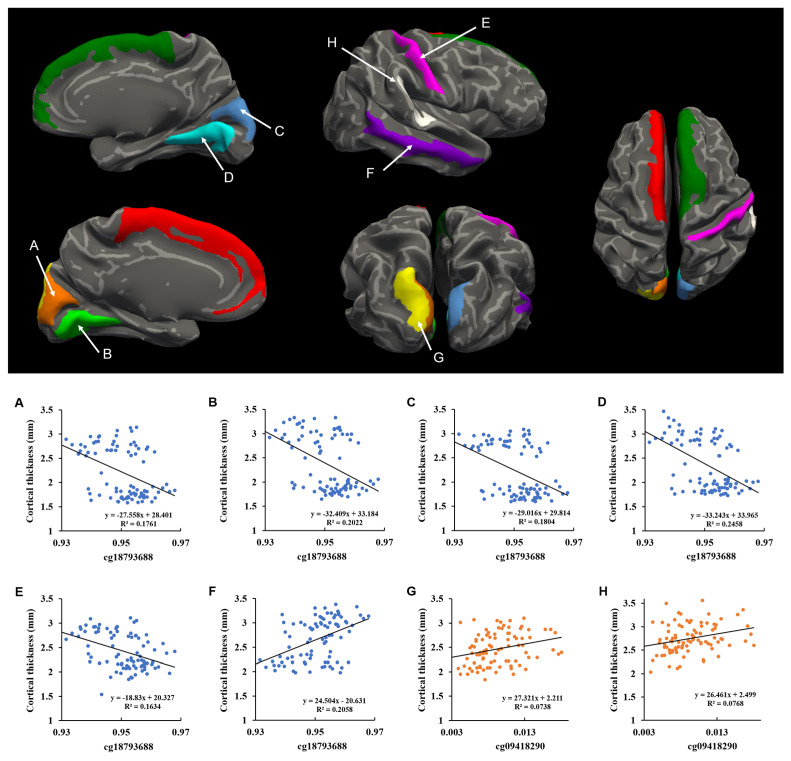
The lower panel shows scatter plots of the correlation analyses between cortical thickness and methylation scores of cg18793688 (**A**–**F**) and cg09418290 (**G**–**H**) in the MDD group on the cortical regions, which showed significant thickness differences between the MDD and HC groups ((**A**)—left cuneus; (**B**)—left lingual gyrus; (**C**)—right cuneus; (**D**)—right lingual gyrus; (**E**)—right postcentral gyrus; (**F**)—right middle temporal gyrus; (**G**)—left superior occipital gyrus; (**H**)—right planum temporale). The upper panel shows schematic maps based on the Destrieux atlas regarding the cortical regions, with significant correlations between their thickness and the methylation scores of cg18793688 and cg09418290 in MDD group. The letters (**A**–**H**) represent the corresponding cortical regions in the scatter plots.

**Table 1 ijms-23-05768-t001:** Demographic and clinical characteristics of MDD patients and HCs included in the neuroimaging analysis.

Characteristics	MDD (n = 88)	HC (n = 74)	*p*-Value (t, χ^2^)
Age	44.38 ± 13.76	30.46 ± 12.06	<0.001 (t = 6.858)
Sex (F/M)	62/26	43/31	0.101 (χ^2^ = 2.687)
Education level			
Elementary and middle school	26	3	<0.001 (χ^2^ = 17.792)
High school or college/university	57	66	
Above graduate school	4	5	
HDRS-17 score	15.47 ± 6.83	1.15 ± 1.90	<0.001 (t = 18.807)
Duration of illness (months)	28.95 ± 45.16	NA	NA
Drug-naïve/Medicated patients	32/56	NA	NA
Remitted/Non-remitted patients	13/75	NA	NA
Medication, n			
SSRI	22	NA	NA
SNRI	13		
NDRI	3		
NaSSA	4		
Other AD	3		
Combination of ADs	11		
AP	12		
Combination of APs	3		

Data are mean ± standard deviation for age, HDRS-17 scores, and illness duration. *p*-values for distribution of sex and education level were obtained using a chi-squared test. *p*-values for comparisons of age and HDRS scores were obtained using an independent *t*-test. Abbreviations: MDD, major depressive disorder; HCs, healthy controls; HDRS-17, 17-item Hamilton Depression Rating Scale; SSRI, selective serotonin reuptake inhibitor; SNRI, serotonin and norepinephrine reuptake inhibitor; NDRI, norepinephrine and dopamine reuptake inhibitor; NaSSA, noradrenergic and specific serotonergic antidepressant; Combination of ADs, combinations of two types of antidepressants; APs, antipsychotics; ADs, antidepressants; Combination of APs, combinations of two types of antipsychotics; NA, not applicable.

**Table 2 ijms-23-05768-t002:** Differentially methylated positions from methylation analysis in the total sample (P_uncorr_ < 0.05).

CpG Site	P_uncorr_	P_corr_	Δβ	Chromosome	Position	Gene	Genomic Feature
cg06710101	0.002	0.015	0.007	1	247587253	*NLRP3*	Body
cg09418290	0.004	0.027	0.001	1	247579319	*NLRP3*	TSS200
cg05615449	0.009	0.049	−0.028	1	247601478	*NLRP3*	Body
cg18126557	0.023	0.091	0.012	1	247611842	*NLRP3*	3′UTR
cg18793688	0.038	0.126	−0.002	1	247588074	*NLRP3*	Body

Abbreviations: P_uncorr_, uncorrected *p*-value; P_corr_, corrected *p*-value; BH (Benjamini–Hochberg)-adjusted *p*-value; Δβ, change in average methylation; 3′ UTR, 3′ untranslated region; Body, gene body; TSS, transcription start site (200—up to 200 bp upstream from TSS). Annotation based on the UCSC (Genome. ucsc. edugenome.ucsc.edu) GRCh37/hg19 reference.

**Table 3 ijms-23-05768-t003:** Cortical thickness comparisons between patients with MDD and HCs.

Cortical Regions	MDD		HC		F	P_uncorr_	P_corr_
	Mean	SD	Mean	SD			
*MDD* < *HC*							
L Ventral posterior cingulate gyrus	2.15	0.44	2.45	0.37	21.116	8.87 × 10^−6^	0.001
L Middle occipital gyrus	2.38	0.27	2.55	0.22	13.491	3.29 × 10^−4^	0.025
L Lateral occipitotemporal gyrus	2.40	0.48	2.77	0.30	33.403	3.95 × 10^−8^	3.01 × 10^−6^
L Superior parietal lobule	2.27	0.22	2.45	0.19	20.706	1.07 × 10^−5^	0.001
L Planum polare	2.96	0.50	3.33	0.39	25.519	1.21 × 10^−6^	9.20 × 10^−5^
L Middle temporal gyrus	2.67	0.45	2.96	0.35	20.095	1.42 × 10^−5^	0.001
R Ventral posterior cingulate gyrus	2.21	0.44	2.69	0.33	48.054	1.03 × 10^−10^	7.85 × 10^−9^
R Short insular gyrus	2.98	0.40	3.26	0.60	12.822	4.57 × 10^−4^	0.035
R Middle occipital gyrus	2.45	0.27	2.60	0.20	13.169	3.85 × 10^−4^	0.029
R Lateral occipitotemporal gyrus	2.44	0.41	2.81	0.28	35.612	1.56 × 10^−8^	1.19 × 10^−6^
R Superior parietal lobule	2.28	0.23	2.43	0.17	16.383	8.13 × 10^−5^	0.006
R Precentral gyrus	2.66	0.23	2.80	0.23	12.840	4.53 × 10^−4^	0.034
R Subcallosal gyrus	2.37	0.42	2.70	0.50	14.193	2.33 × 10^−4^	0.018
R Planum polare	2.87	0.44	3.16	0.38	15.408	1.30 × 10^−4^	0.010
R Middle temporal gyrus	2.67	0.43	2.92	0.30	17.928	3.91 × 10^−5^	0.003
***MDD* > *HC***							
L Paracentral lobule	2.52	0.23	2.44	0.18	15.162	1.46 × 10^−4^	0.011
L Posterior mid-cingulate gyrus	2.68	0.19	2.56	0.23	23.786	2.63 × 10^−6^	2.00 × 10^−4^
L Cuneus	2.20	0.52	1.94	0.39	22.871	3.98 × 10^−6^	3.02 × 10^−4^
L Superior occipital gyrus	2.47	0.33	2.12	0.26	61.294	7.03 × 10^−13^	5.34 × 10^−11^
L Lingual gyrus	2.37	0.57	2.03	0.30	30.837	1.18 × 10^−7^	8.98 × 10^−6^
L Postcentral gyrus	2.43	0.34	2.22	0.19	33.786	3.36 × 10^−8^	2.56 × 10^−6^
L Precuneus	2.64	0.21	2.55	0.19	22.229	5.33 × 10^−6^	4.05 × 10^−4^
L Lateral superior temporal gyrus	3.18	0.30	3.00	0.30	22.420	4.88 × 10^−6^	3.71 × 10^−4^
R Frontomarginal gyrus	2.47	0.26	2.36	0.23	12.997	4.19 × 10^−4^	0.032
R Posterior mid-cingulate gyrus	2.71	0.19	2.59	0.19	24.728	1.72 × 10^−6^	1.31 × 10^−4^
R Cuneus	2.22	0.54	1.95	0.40	23.243	3.36 × 10^−6^	2.56 × 10^−4^
R Superior occipital gyrus	2.47	0.33	2.18	0.22	55.742	5.47 × 10^−12^	4.16 × 10^−10^
R Lingual gyrus	2.35	0.53	2.10	0.29	26.666	7.28 × 10^−7^	5.53 × 10^−5^
R Postcentral gyrus	2.42	0.37	2.22	0.20	27.926	4.18 × 10^−7^	3.18 × 10^−5^
R Lateral superior temporal gyrus	3.14	0.24	2.97	0.36	18.004	3.77 × 10^−5^	0.003
R Planum temporale	2.75	0.32	2.62	0.22	13.815	2.81 × 10^−4^	0.021

Cortical regions with significant difference after Bonferroni correction are presented. Analysis of covariance included covariates of age, sex, education, and total intracranial cavity volume. Bonferroni correction was applied: 0.05/76 = 6.57 × 10^−4^. Abbreviations: MDD, major depressive disorder; HC, healthy controls; P_uncorr_, uncorrected *p*-value; P_corr_, corrected *p*-value; L, left hemisphere; R, right hemisphere; SD, standard deviation.

**Table 4 ijms-23-05768-t004:** Correlation between *NLRP3* DNA methylation and cortical thickness in patients with MDD.

Cortical Regions	cg18793688			cg09418290		
	r	P_uncorr_	P_corr_	r	P_uncorr_	P_corr_
L Cuneus *	−0.364	**8.46 × 10^−4^**	**0.038**	0.270	0.015	0.118
L Superior frontal gyrus	−0.359	**0.001**	**0.038**	0.272	0.014	0.118
L Superior occipital gyrus *	−0.276	0.013	0.118	**0.369**	**6.87 × 10^−4^**	**0.037**
L Lingual gyrus *	−0.383	**4.19 × 10^−4^**	**0.037**	0.273	0.014	0.118
R Cuneus *	−0.378	**5.10 × 10^−4^**	**0.037**	0.261	0.019	0.140
R Superior frontal gyrus	−0.388	**3.42 × 10^−4^**	**0.037**	0.238	0.033	0.191
R Lingual gyrus *	−0.447	**2.94 × 10^−5^**	**0.011**	0.289	0.009	0.103
R Postcentral gyrus *	−0.394	**2.72 × 10^−4^**	**0.037**	0.282	0.011	0.110
R Planum temporale *	−0.332	0.002	0.052	**0.373**	**6.08 × 10^−4^**	**0.037**
R Middle temporal gyrus †	0.362	**9.07 × 10^−4^**	**0.038**	−0.254	0.022	0.151

A Pearson’s correlation analysis included covariates of age, sex, education level, total intracranial cavity volume, Hamilton Depression Rating Scale score, illness duration, and medication. The false discovery rate (FDR) was applied in each analysis for multiple-comparison correction, q < 0.05 (76 cortical regions × 5 differentially methylated positions). * Greater cortical thickness in MDD (MDD > HC) is marked with an asterisk. † Lesser cortical thickness in MDD (MDD < HC) is marked with a dagger. Abbreviations: P_uncorr_, uncorrected *p*-value; P_corr_, corrected *p*-value; L, left hemisphere; R, right hemisphere; MDD, major depressive disorder.

## Data Availability

Not applicable.

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
