# Peer review of "Association of DNA Methylation of the NLRP3 Gene with Changes in Cortical Thickness in Major Depressive Disorder"

_ijms, 2022, doi:10.3390/ijms23105768_

Round 1
Reviewer 1 Report
This study in220 MDD patients and 82 HCs explores the differential methylation pattern in the Nod-like receptor pyrin containing 3 (NLRP3) inflammasome to examine the contribution of epigenetic regulation of NLRP3 to the pathophysiology of MDD. In addition the authors investigated the correlationof methylation score of loci in NLRP3 to cortical thickness in the MDD group using T1- weighted magnetic resonance imaging (MRI) data to elucidate the relationship between epigenetic regulation of NLRP3 and structural brain changes in MDD. After comprehensive quality control procedures, 753,531 CpGs were retained and 105 used for DMP analysis. They identified five significant DMPs (P < 0.05, Q = 0.15) in the 106 NLRP3 gene, which included DMPs with absolute changes in average methylation (∆β) 107 ranging from 0.001 to 0.028. Their correlation analysis using the methylation score data of five significant DMPs and cortical thickness from the neuroimaging sample showed that the methylation scores of cg18793688 and cg09418290 in NLRP3 were 140 significantly correlated with cortical thickness in MDD patients.
I have a series of comments:
- it is not clear to me why the authors make the effort of doing an epigenomic analysis to focus only on NLRP3
- How was the gene prioritized in the analysis?
- There is a wealth of clinical data but just minima information was used for the analysis (sex and age)
- It is unclear why the authors did not test for the impact of pharmacological therapy
- Some permutation analysis would help with the multiple comparison issue
Author Response
Reviewer 1
This study in 220 MDD patients and 82 HCs explores the differential methylation pattern in the Nod-like receptor pyrin containing 3 (NLRP3) inflammasome to examine the contribution of epigenetic regulation of NLRP3 to the pathophysiology of MDD. In addition the authors investigated the correlation of methylation score of loci in NLRP3 to cortical thickness in the MDD group using T1- weighted magnetic resonance imaging (MRI) data to elucidate the relationship between epigenetic regulation of NLRP3 and structural brain changes in MDD. After comprehensive quality control procedures, 753,531 CpGs were retained and 105 used for DMP analysis. They identified five significant DMPs (P < 0.05, Q = 0.15) in the 106 NLRP3 gene, which included DMPs with absolute changes in average methylation (∆β) 107 ranging from 0.001 to 0.028. Their correlation analysis using the methylation score data of five significant DMPs and cortical thickness from the neuroimaging sample showed that the methylation scores of cg18793688 and cg09418290 in NLRP3 were 140 significantly correlated with cortical thickness in MDD patients.
I have a series of comments:
it is not clear to me why the authors make the effort of doing an epigenomic analysis to focus only on NLRP3
Response: Thank you for your thorough review of our manuscript. We believe that your suggestions have substantially helped us improve it. Our point-by-point responses to your comments are provided below.
The NLRP3 inflammasome has been suggested to be deeply involved in the neuroinflammation-related pathophysiology of mood disorders, including MDD (Bhattacharya & Jones, 2018; Kaufmann et al., 2017). A growing body of evidence has shown that DNA methylation of the NLRP3 gene plays a critical role in expression, assembly, and activation of the NLRP3 inflammasome (Raneros et al., 2021). We aimed to examine whether epigenetic regulation of the NLRP3 gene is associated with the diagnosis of MDD and/or MDD-related brain structural changes. Thus, we performed an epigenomic analysis focusing on the NLRP3 gene. As per the reviewer’s suggestion, we have added the following text explaining why the authors focused on NLRP3, in the Introduction section:
“Furthermore, a growing number of studies have shown that DNA methylation of the NLRP3 gene plays a critical role in expression, assembly, and activation of the NLRP3 inflammasome [3].”
“To examine our a priori hypothesis, we performed an epigenomic analysis focusing on the NLRP3 gene and cortical thickness analysis.”
Bhattacharya, A.; Jones, D. N. C. Emerging role of the P2X7-NLRP3-IL1β pathway in mood disorders. Psychoneuroendocrinology 2018, 98, 95–100. doi:10.1016/j.psyneuen.2018.08.015
Kaufmann, F. N.; Costa, A. P.; Ghisleni, G.; Diaz, A. P.; Rodrigues, A. L. S.; Peluffo, H.; Kaster, M. P. NLRP3 inflammasome-driven pathways in depression: clinical and preclinical findings. Brain Behav. Immun. 2017, 64, 367–383. doi:10.1016/j.bbi.2017.03.002
Raneros, A. B.; Bernet, C. R.; Flórez, A. B.; Suarez-Alvarez, B. An epigenetic insight into NLRP3 inflammasome activation in inflammation-related processes. Biomedicines 2021, 9(11). doi:10.3390/biomedicines9111614
How was the gene prioritized in the analysis?
Response: Several inflammatory pathways and neurocircuits in the brain are associated with MDD and the response to antidepressant treatment (Miller & Raison, 2016). Recent studies have reported the mechanisms by which innate and adaptive immune systems interact with neurotransmitters and neurocircuits to influence the risk of MDD (Miller & Raison, 2016; Pan et al., 2014). All these mechanisms suggest the NLRP3 inflammasome as a biologically plausible target for the prevention and treatment of MDD (Paugh et al., 2015; Zhang et al., 2015). Therefore, based on a comprehensive systematic literature review, we hypothesized that the NLRP3 inflammasome plays a critical role in neuroinflammation-related pathophysiological processes in MDD. We then performed an epigenomic analysis focusing on the NLRP3 gene based on our a priori hypothesis that DNA methylation of the NLRP3 gene, which plays an important role in expression, assembly, and activation of the NLRP3 inflammasome, may be associated with the diagnosis of MDD and MDD-related brain structural alterations (i.e., cortical thickness).
Miller, A. H.; Raison, C. L. The role of inflammation in depression: from evolutionary imperative to modern treatment target. Nat. Rev. Immunol. 2016, 16, 22–34.
Pan, Y.; Chen, X.-Y.; Zhang, Q.-Y.; Kong, L.-D. Microglial NLRP3 inflammasome activation mediates IL-1β-related inflammation in prefrontal cortex of depressive rats. Brain. Behav. Immun. 2014, 41, 90–100.
Zhang, Y. et al. NLRP3 inflammasome mediates chronic mild stress-induced depression in mice via neuroinflammation. Int. J. Neuropsychopharmacol. 2015, 18, pyv006.
Paugh, S. W. et al. NALP3 inflammasome upregulation and CASP1 cleavage of the glucocorticoid receptor cause glucocorticoid resistance in leukemia cells. Nat. Genet. 2015, 47, 607–614.
There is a wealth of clinical data but just minimal information was used for the analysis (sex and age).
Response: As the reviewer commented, it is important to consider the influence of the demographic and clinical information. In the present study, for the DNA methylation-neuroimaging correlation analysis in the MDD group, the variables of education level, total intracranial cavity volume (TICV), illness duration, depression severity (measured by the 17-item Hamilton Depression Rating Scale), and psychotropic medication (drug-naïve patients, coded as 0; patients taking psychotropic medications, coded as 1) were included as covariates in the partial correlation analysis to minimize the impact of these confounding factors. For DNA methylation analysis, we only included age and sex as covariates, according to previous DNA methylation studies on psychiatric disorders (Zhou et al., 2021). In a review article of 39 studies on DNA methylation changes associated with schizophrenia, bipolar disorder, and MDD, the authors observed that nearly 49% of the studies did not control for covariates in their analyses, and when there was a correction for covariates, only sex and age were controlled (Zhou et al., 2021). However, other demographic factors such as body mass index, smoking, and alcohol consumption may affect the results of DNA methylation analysis (Zhou et al., 2021). We have added the following comment in the limitations section:
“Finally, we did not include variables such as body mass index, cigarette smoking, or alcohol consumption, which can affect DNA methylation status [49], in the DNA methylation analysis, which may have affected our results.”
Zhou, J.; Li, M.; Wang, X.; He, Y.; Xia, Y.; Sweeney, J. A.; . . . Chen, C. Drug response-related DNA methylation changes in schizophrenia, bipolar disorder, and major depressive disorder. Front. Neurosci. 2021, 15, 674273. doi:10.3389/fnins.2021.674273
It is unclear why the authors did not test for the impact of pharmacological therapy
Response: We could not find any potential impact of the variable of taking psychotropic medication (drug-naïve patients, coded as 0; patients taking psychotropic medications, coded as 1) on DNA methylation and neuroimaging analysis. We have added the following comments to the Results section:
“We did not observe any significant association between the variable of taking psychotropic medication and the methylation scores of five significant DMPs. Furthermore, there was no significant difference in cortical thickness between drug-naïve patients and those taking psychotropic medications in the MDD group.”
Some permutation analysis would help with the multiple comparison issue
Response: To correct for multiple testing, we used the false discovery rate (FDR) developed by Benjamini and Hochberg (Benjamini & Hochberg, 1995) for both DNA methylation and neuroimaging analyses, according to previous epigenetic studies in MDD (Barbu et al., 2021; Humphreys et al., 2019; Lapato et al., 2019).
Barbu, M. C.; Shen, X.; Walker, R. M.; Howard, D. M.; Evans, K. L.; Whalley, H. C.; McIntosh, A. M. Epigenetic prediction of major depressive disorders. Mol. Psychiatry 2021, 26(9), 5112–5123. doi:10.1038/s41380-020-0808-3
Benjamini, Y.; Hochberg, Y. Controlling the false discovery rate: a practical and powerful approach to multiple testing. J. R. Stat. Soc. Series B (Methodological) 1995, 57(1), 289–300.
Humphreys, K. L.; Moore, S. R.; Davis, E. G.; MacIsaac, J. L.; Lin, D. T. S.; Kobor, M. S.; Gotlib, I. H. DNA methylation of HPA axis genes and the onset of major depressive disorder in adolescent girls: A prospective analysis. Transl. Psychiatry 2019, 9(1), 245. doi:10.1038/s41398-019-0582-7
Lapato, D. M.; Roberson-Nay, R.; Kirkpatrick, R. M.; Webb, B. T.; York, T. P.; Kinser, P. A. DNA methylation associated with postpartum depressive symptoms overlaps findings from a genome-wide association meta-analysis of depression. Clin. Epigenetics 2019, 11(1), 169. doi:10.1186/s13148-019-0769-z
Reviewer 2 Report
This is most interesting contribution which explores the DNA methylation of NLRP3 and cortical thickness by means of magnetic resonance imaging in patients with MDD. The structure of the paper needs improvement.
The material and methods section from lines 288 to 380 needs to be replaced after the introduction and before the results section on line 88.
There is a significant difference in the sample with both DNA methylation status and neuroimaging data collected and the sample with DNA methylation analysis only. The clinical and demographic characteristics of the two samples as well as the results should be either reported separately within this paper or reported in two separate papers, to avoid confusion.
Other then those technical concerns the paper may benefit from more careful discussion on the role of epigenetics in clinical neuropsychiatry (e.g., not limited to https://link.springer.com/article/10.1007/s13148-010-0014-2), and previous studies on the reduction of gray matter volume in MDD with voxel-based neuro-morphometry (e.g. not limited to: https://pubmed.ncbi.nlm.nih.gov/31234950/)
Author Response
Review 2
This is most interesting contribution which explores the DNA methylation of NLRP3 and cortical thickness by means of magnetic resonance imaging in patients with MDD.
The structure of the paper needs improvement.
Response: Thank you for your thorough review of our manuscript. We believe that your suggestions have substantially helped us improve it. Our point-by-point responses to your comments are provided below.
The material and methods section from lines 288 to 380 needs to be replaced after the introduction and before the results section on line 88.
Response: This journal (i.e., IJMS) provides the format of an original article in the order of “Introduction-Results-Discussion-Methods/Materials. The authors followed the policies of this journal.
There is a significant difference in the sample with both DNA methylation status and neuroimaging data collected and the sample with DNA methylation analysis only. The clinical and demographic characteristics of the two samples as well as the results should be either reported separately within this paper or reported in two separate papers, to avoid confusion.
Response: In the present study, we designed two-step analyses based on our previous imaging-genetic studies combining whole-exome sequencing and MRI data (K. M. Han et al., 2020; M. R. Han et al., 2019): First, DNA methylation analysis was performed on the total sample (i.e., MDD, n = 220; HC, n = 88), and we examined whether there was any correlation between DNA methylation status (i.e., differentially methylated positions in the NLRP3 gene) and neuroimaging markers in sub-samples consisting of patients who underwent T1 MR scans. This is because only a few participants underwent brain MRI. As per the reviewer’s suggestion, we have added a supplemental table (i.e., Table S1) describing detailed information about the demographic and clinical characteristics of the participants in the total sample.
Han, K. M.; Han, M. R.; Kim, A.; Kang, W.; Kang, Y.; Kang, J.; . . . Ham, B. J. A study combining whole-exome sequencing and structural neuroimaging analysis for major depressive disorder. J. Affect Disord. 2020, 262, 31–39. doi:10.1016/j.jad.2019.10.039
Han, M. R.; Han, K. M.; Kim, A.; Kang, W.; Kang, Y.; Kang, J.; . . . Ham, B. J. Whole-exome sequencing identifies variants associated with structural MRI markers in patients with bipolar disorders. J. Affect Disord. 2019, 249, 159–168. doi:10.1016/j.jad.2019.02.028
Other then those technical concerns the paper may benefit from more careful discussion on the role of epigenetics in clinical neuropsychiatry (e.g., not limited to https://link.springer.com/article/10.1007/s13148-010-0014-2), and previous studies on the reduction of gray matter volume in MDD with voxel-based neuro-morphometry (e.g. not limited to: https://pubmed.ncbi.nlm.nih.gov/31234950/)
Response: We would like to express our deep gratitude to the reviewers for mentioning these valuable studies. As per the reviewer’s suggestion, we have added the following two comments regarding the 1) important role of epigenetics in the pathophysiology of MDD and 2) gray-matter volume reduction in MDD based on the suggested references, in the Discussion section.
“A growing body of evidence has shown that methylation signatures can provide a deep insight into the interplay between psychosocial environment and disease trajectory of MDD [11, 12]. Our findings suggest that the neuroinflammation-related methylomic profiles of individuals may provide useful information about the neurobiological process among psychosocial stress, heightened inflammation states, structural alterations of the brain, and predisposition to MDD [2, 49, 50]. This could be helpful in the realization of the concept of ‘evidence-based psychiatry’ in MDD [51]. Future studies are needed to confirm the utility of DNA methylation of the NLRP3 gene as a biomarker for neuroinflammatory changes and pathophysiological processes in MDD.”
“Particularly, cortical thinning in the right middle frontal gyrus, which has a positive correlation with the cg18793688 methylation score, was inconsistent with that in a previous voxel-based morphometry study, which reported gray-matter volume reduction in the middle temporal gyrus and PFC in the right hemisphere in MDD compared to that in HCs [33].”
Round 2
Reviewer 2 Report
In my understanding a clinico-biological research paper should present participants/methods first, and then proceed to the results section. This is the standard structure in other MDPI journals as well.
The Editorial office is advised to provide an exception for this publication to deviate from the IJMS format, as it is not appropriate for a clinical article.